# Blockchain-Based Smart Farm Security Framework for the Internet of Things

**DOI:** 10.3390/s23187992

**Published:** 2023-09-20

**Authors:** Ahmed Abubakar Aliyu, Jinshuo Liu

**Affiliations:** 1School of Cyber Science and Engineering, Wuhan University, Wuhan 430072, China; liujinshuo@whu.edu.cn; 2Department of Computer Science, Faculty of Computing, Kaduna State University, Kaduna 800283, Nigeria

**Keywords:** blockchain, poisoning attacks, Internet of Things, smart farming

## Abstract

Smart farming, as a branch of the Internet of Things (IoT), combines the recognition of agricultural economic competencies and the progress of data and information collected from connected devices with statistical analysis to characterize the essentials of the assimilated information, allowing farmers to make intelligent conclusions that will maximize the harvest benefit. However, the integration of advanced technologies requires the adoption of high-tech security approaches. In this paper, we present a framework that promises to enhance the security and privacy of smart farms by leveraging the decentralized nature of blockchain technology. The framework stores and manages data acquired from IoT devices installed in smart farms using a distributed ledger architecture, which provides secure and tamper-proof data storage and ensures the integrity and validity of the data. The study uses the AWS cloud, ESP32, the smart farm security monitoring framework, and the Ethereum Rinkeby smart contract mechanism, which enables the automated execution of pre-defined rules and regulations. As a result of a proof-of-concept implementation, the system can detect and respond to security threats in real time, and the results illustrate its usefulness in improving the security of smart farms. The number of accepted blockchain transactions on smart farming requests fell from 189,000 to 109,450 after carrying out the first three tests while the next three testing phases showed a rise in the number of blockchain transactions accepted on smart farming requests from 176,000 to 290,786. We further observed that the lesser the time taken to induce the device alarm, the higher the number of blockchain transactions accepted on smart farming requests, which demonstrates the efficacy of blockchain-based poisoning attack mitigation in smart farming.

## 1. Introduction

Normally, as the population of the world grows, so does our need for agricultural improvement. Farmers work to produce crops that will provide food for people all over the world as the economies of most countries are primarily dependent on the agricultural industry [1]. Moreover, many nations have agricultural departments that work to strengthen their country’s economy, especially through agriculture. Over the past few decades, it has become clear that the growth of the IoT has revolutionized the way farming is done and advanced the operational capabilities of the agricultural sector [2,3]. The integration of the IoT into agricultural growth is known as smart farming, and it is quickly fitting in as the new normal as connected devices, smart things, and robots exhibited around the globe are expected to be valued at around $15.93 billion in 2028, representing a yearly growth ratio of about 20.31% between 2021 and 2028 [4]. Similarly, modern agricultural frameworks are integrated into rural regions and competitors are targeting them for cyber-attacks. For example, a ransomware outbreak at the food transportation division of the meat management company JBS halted operations at 13 meat industrial sites. To remain operational, the company had to spend approximately $11 million [5]. As a result, we can all agree that security is seen as a key concern in industries such as agriculture, where the advancement of rural security measures is vital.

Poisoning attacks are a form of cyber-attack that targets the data used by IoT devices [6]. In order to damage the system or take over the devices, the attacker injects harmful or false data into the system. Data poisoning and sensor poisoning are the two basic forms of poisoning attacks. Data poisoning involves changing the data used to train or calibrate an IoT device. This can be done by introducing false data into the system or by manipulating existing data. Data poisoning aims to cause the device to generate false results. Sensor poisoning involves tampering with the sensors used by an IoT device. This can be done by physically altering the sensors or by hacking into the sensors’ software. The purpose of sensor poisoning is to cause the device to collect false data. Attacks involving poisoning have a major impact on IoT systems [7]. They can cause devices to malfunction, produce inaccurate data, or even allow the attacker to take control of the device.

The cybersecurity structures now advocated in smart farming typically include chain management of food supply and testing of several accomplishments through data analysis techniques, cloud computing technologies as well as verification and authorization arrangements for sophisticated IoT devices based on machine learning/artificial intelligence [8,9]. It has also been observed that real IoT devices identified on the Internet were infiltrated and employed as a means to launch full denial-of-service (DoS) assaults and further harmful engagements, such as information leakage related to management and sensor data [10]. On the other hand, blockchain has emerged and evolved in a fascinating way and is currently being used in decentralized network systems such as IoTs [11]. Researchers have made a separate assessment of the blockchain advancement gaps for IoT security and safety difficulties, and they have advised and urged us to use blockchain-based monitoring for the general security of smart agriculture [12]. Traditionally, in order to extend the limitations of the current system and make progress in terms of security with blockchain-based system needs, we use blockchain arrangements to continuously handle information and store irregularities in blockchain transactions [13,14]. In this study, the AWS cloud, an Arduino gadget package with a Wi-Fi component (in Figure 1), and an Ethereum smart contract were used as an end-to-end action.

Given that blockchain technology has the potential to revolutionize smart agriculture [16] by improving transparency, traceability, and efficiency in the agricultural supply chain, it is important to note that the suitability of a particular blockchain technology for smart agriculture depends on the specific requirements, scalability needs, and existing infrastructure of the agricultural system [17]. In order to select the best platform, it is important to weigh the advantages and disadvantages of each. Smart farming often uses a variety of blockchain technologies, including:Ethereum: Ethereum is one of the most popular blockchain platforms for decentralized applications (dApps) and smart contracts [18] as it provides a stable and adaptable environment for creating agricultural blockchain solutions. Ethereum’s default cryptocurrency, Ether (ETH), enables secure and open transactions across the entire ecosystem. In addition, the scalability issues it faces, particularly high transaction costs and slow processing times, could limit the use of smart agriculture in high-volume environments [19].Hyperledger Fabric: Hyperledger Fabric is an open source, enterprise-grade blockchain platform with a modular design that gives designers and developers more freedom to create and implement smart farming solutions [20]. Fabric uses a permissioned network, giving users and collaborating companies limited access and privacy. It also includes pluggable consensus processes that allow for modification based on specific use cases and is focused on enterprise solutions, making it suitable for widespread smart agriculture installations.Corda: Corda is a distributed ledger platform that emphasizes privacy and security by limiting data access to only participating parties. Corda is designed for commercial applications using “CorDapps”, smart contracts that enable secure and direct transactions throughout the agricultural supply chain [21]. Without exposing transaction information to the whole network, its original “notary” approach ensures consensus. Corda’s emphasis on privacy and facilitating direct interaction makes it suitable for sensitive and challenging smart farming environments.IBM Food Trust: IBM Food Trust is a blockchain-based platform designed specifically for the food sector, including agriculture. It enables end-to-end traceability of food, providing accountability and transparency. It also combines the benefits of enterprise-grade functionality and permissioned networks, using Hyperledger Fabric as the underlying blockchain technology. As a result, farmers, distributors, retailers, and consumers can use the platform to access reliable information about the origin and movement of their food. The quality and reliability of the information shared is enhanced by IBM Food Trust’s integration with multiple data sources, such as IoT devices and sensors [22].VeChain: VeChain is a blockchain platform focused on product authenticity and supply chain management. It uses a two-token structure, with VET as the native coin and VTHO as the fee and smart contract execution token. Throughout the supply chain, VeChain provides features such as Near Field Communication (NFC) chips and QR codes to track and validate agricultural products. In addition, its ecology and adoption potential are strengthened by its links with multiple businesses and government organizations. VeChain’s focus on product authenticity and supply chain management makes it ideal for ensuring food safety and quality in smart agriculture [23].

This study is expected to have a promising significance to farmers, the government, and also cybersecurity and assurance specialists as it renders various scenarios of data and information attacks that have been encountered by smart farm administrators globally. The research is also aimed at recognizing possible cybersecurity alarms in smart farming and presenting scenario-specific cyber-attacks. It also intends to provide a comprehensive evaluation of current cybersecurity analyses, as well as present a preventive measure through a blockchain technology consensus in an intelligent farming ecosystem.

Data breaches caused by improperly set up access controls are a security concern associated with cloud-based data storage. Blockchain, however, makes it possible to store documents securely without spending money on storage, whereas our scalable cloud approach provides answers to a number of smart farm security use cases. In addition to ensuring the security of insurance claims and data-corruption-free security investigations to secure farm assets, the immutability of blockchain transactional alarm data can be used as evidence in court cases. For example, natural disasters can have a significant impact on agricultural land, and transaction data can be used to store evidence of the when, what, where, and how, which can then be used in insurance claims. Once a transaction is recorded on the blockchain, a farm can no longer claim ownership. Moreover, sensors continuously track the physical health of farms and transmit this information so that farmers can optimize their operations to improve yields, reduce losses, and increase productivity. The health of the sensors on these devices must be monitored regularly, as they are vulnerable to passive and active attacks. A mobile application must also alert the farmer when the health of the device is compromised. The farmer can then identify the root cause of the problem and fix it.

## 2. Literature Review

Light was shed [24] on security issues in the IoT in general and smart agriculture in particular, where the production of coatings and notable conceivable cyber threats in smart agriculture were presented. Additionally, their research provides certain cyber-attack situations characterized by data, features, and other attacks. A predominant attack called “The Night Mythical Serpent” is a framework that allows network intruders to get huge amounts of data from several petrochemical corporations. The growing number of connected devices has created a lot of safety and security challenges within the smart farming ecosystem in rural areas because farmers cannot endure severe damages to their crops. Maria and partners’ [25] report, where they explained dangers and potential vulnerabilities in the emerging IoT terrain, highlights the importance of data security in smart farming. Their research focused on security, intelligence, and accessibility models for information security in agriculture, as well as unique advances in smart farm systems, such as on-farm equipment verification, inaccessible sensing approaches, and machine learning. Moreover, the risks associated with the use of IoT technology in agriculture have been clearly identified [26].

Recently, an expert from the security firm Sucuri [27] discovered that a DoS botnet may send 50,000 HTTP requests per second, causing DDoS attacks on many domains. Cloud computing’s integration with Smart Farming is critical for establishing IoT identifying information capacity and analysis, as well as tallying big data demands. Thus, researchers proposed strategies for solving IoT-based Smart Farming problems using cloud computing [28]. Furthermore, a research paper proposed a blockchain-based device management system for smart city security considerations, using a private blockchain to verify device integrity and record results. It provides four end node management protocols, including two for heavy and light end nodes. Also, the framework provides bi-directional update protocols, device firmware monitoring, information sharing, and security threat response. The architecture has the potential to dramatically increase network availability and security, potentially extending to high-reliability applications [29]. Similarly, a comprehensive authentication method for IoT devices using CoAP is proposed in another study as the Cyber Secured Framework for Smart Agriculture (CSFSA), which ensures the authenticity and integrity of data [30]. The CSFSA is effective for memory-constrained systems and robust against resource exhaustion and cybersecurity attacks. It can significantly reduce both food waste and financial loss.

Recent research [31] has proposed an architecture that provides efficient, secure data exchange in a distributed environment using a blockchain-based IoT data communication system with an event-driven smart contract. With IoT device connection setup and client subscription taking only a few seconds, the Ethereum-based simulation tool enables testing in different IoT configurations. The architecture is a good option for IoT devices with limited resources, as it provides a consistent data connection with low latency and resource consumption. A paper [32] presented SP2F, a secure, privacy-preserving integrated framework which blends deep learning and blockchain technologies for smart farming. The framework has a privacy engine with two layers: the first layer uses a blockchain and smart contract-based enhanced proof of work (ePoW) and the second layer uses the sparse autoencorder (SAE) technique to transform data into a new encrypted format.

A lot of experts have explored the use of blockchain technology for IoT advancement owing to the several benefits it provides, which include green computing [10,33,34]. Cloud solutions in smart farming refer to the use of cloud computing technology to enhance farming operations and improve crop yields. Also, cloud solutions enable farmers to accumulate, store, and evaluate data from multiple sources, such as drones, soil sensors, and weather sensors, and then use this information to make data-driven choices about pest control, irrigation, and fertilization. With cloud solutions, farmers can access real-time data from anywhere and use it to optimize their farming practices and increase productivity. In addition, cloud solutions can help farmers reduce costs and minimize waste by providing accurate predictions of crop yields and enabling them to fine-tune their operations accordingly. Overall, cloud solutions are becoming an increasingly important part of modern agriculture, helping farmers achieve greater efficiency, sustainability, and profitability. Figure 2 illustrates smart farming applications in cloud-based IoT while Table 1 compares the related work in the literature.

## 3. Methodology

The proposed methodology aims to improve the security and monitoring of the smart agriculture system. The Ethereum blockchain is used to track smart contracts and trigger events when discrepancies in security checks are detected. Figure 2 shows the layered design of the proposed approach. These IoT devices continuously generate events, such as device status, device information, and so on. The generated events are sent to the cloud through a wireless gateway or switch connected to the device. The cloud layer consists of components that continuously monitor the device events and process the event data to extract the required data in the system. MQTT is the industry standard for end-to-end packet data transmission. In the AWS cloud, we developed a Lambda function to analyze data from the AWS IoT main component and extricate the relevant data from sensor devices attached to the farms. When the Lambda identifies a security warning in the device data generation, it initiates an Infura API POST request to update the Ethereum blockchain. Moreover, the improved exchange may include anomalous values of device information, device location, etc. Infura operates Ethereum hubs and provides an API for upgrading ex-variations from customer accounts if they have one. Also, upgraded blockchain ex-variations will be made available on all Ethereum hubs. Though Figure 3 did not illustrate the client layer, the GUI could examine transactions from the Ethereum hub by means of an API call.

### Components Used in Our Approach

The portrayal of most of the parts utilized within the projected approach is discussed below:Ethereum: Form works on the POS agreement component to favor and incorporate ex-variations to the Ethereum blockchain. When a safety event is detected, a Web3 frontend request is conducted to survey and warn the farmers.Infura API: This is a feature of Ethereum API that allows smart contracts to be performed in Ethereum hubs and performs Ethereum-based ex-variations. Once we have collected and prepared the farming device data, we use the Infura API calls to connect with Ethereum hubs.AWS IoT core: Several IoT devices’ sensors are available in the smart agricultural environment. To gather messages from diverse IoT devices, a message-processing framework is necessary to supplement IoT message protocols such as the MQTT and suit the organized transfer speed. Furthermore, to benefit from the smart agricultural IoT data preparation, we chose the AWS IoT core. The AWS IoT core enables minimal inactivity and maximum throughput execution, which aids in the development of real-time production-level IoT monitoring frameworks.AWS Lambda: The IoT data should be collected, prepared, and sent into the system as input data. As a result, AWS Lambda performs the cryptography in the background and saves the smart farming data to the blockchain. AWS Lambda is a serverless computing service from Amazon Web Services (AWS) that lets you run code without deploying or managing servers. With AWS Lambda, we can write and upload our code in the form of functions, while it takes care of the underlying infrastructure required to run those functions. Some of the key features of AWS Lambda include: serverless architecture, event-driven execution, broad language support, automatic scaling, integration with AWS services, easy deployment and management, and pay-per-use pricing. AWS Lambda provides a flexible and scalable way to execute code without worrying about infrastructure management as it is widely used for building serverless applications, event-driven architectures, and implementing various backend tasks in the AWS ecosystem [35]. Figure 4 shows the security framework activity diagram.

## 4. Implementation and Results Discussion

To quickly inform the farmer of a problem, we trigger a quick device alarm. In certain situations, it is critical to notify the user of a problem as soon as it occurs. This is especially true for safety-critical devices such as IoT industrial and medical equipment. With a quick alert, the user can be made aware of the problem and to act to mitigate it before it gets worse. First, we need to trigger a device alarm based on the organized sleep in seconds and check the number of blockchain transactions accepted for smart farming requests. We then set up a device alarm to determine the conditions that should trigger the device alarm based on the organized sleep in seconds. For example, if the sleep time exceeds a certain threshold, we trigger the alarm and then use the appropriate hardware or software components to generate the alarm signal, such as a buzzer, notification, or API call. Then we track the organized sleep time in seconds. This could involve using a timer or timestamp mechanism within our code or application that monitors the sleep period and keeps track of its duration. Next, we monitor the blockchain transactions by integrating a solution that allows us to monitor blockchain transactions related to smart farm requests. This could involve interacting directly with the blockchain network or using APIs provided by the blockchain platform or service we are using. In addition, we implement logic within our application to count the number of blockchain transactions accepted for smart farm requests. This may involve querying the blockchain for relevant transactions, filtering based on certain criteria (such as transaction types or smart contract interactions), and incrementing a counter for each accepted transaction. Finally, if the organized sleep duration exceeds the specified threshold and the number of accepted blockchain transactions meets our audit criteria, we trigger the device alarm. This could involve invoking the alarm mechanism we set up in step 1, such as sounding the buzzer, sending a notification, or making an API call to our device control system. Figure 5 and Figure 6 show alerts for smart-contract web applications’ frontend and the frontend GUI for smart-contract web application.

Our data were obtained from the device alarm triggering phase, when organized sleep is in seconds, to verify the number of blockchain transactions accepted based on smart farming requests. Organized sleep is a period of time when the blockchain network does not accept new transactions. This can be done to conserve energy or to prevent the network from becoming overloaded. The specific length of organized sleep depends on a number of factors, such as the number of devices connected to the network, the amount of data being transferred, and the desired level of security. Organized idleness also improves the efficiency of blockchain transactions by coordinating the idle time of devices on the network. This can be achieved by a software agent that monitors the network and aggregates idle devices for transaction processing. A blockchain-based system rewards devices for their idle time, making them more available for transaction processing. The data could be used to determine the effectiveness of the blockchain-based smart farming system. For example, if the number of accepted blockchain transactions is high, this could indicate that the system is working well and that farmers are able to use it to manage their farms efficiently. The data could also be used to identify any problems with the system. For example, if the number of blockchain transactions accepted is low, it could indicate that there is a problem with the system, such as a lack of connectivity or a security breach. Our tests were carried out in six phases as shown in Table 2, and the data obtained are shown in Section 4.1.

### 4.1. Test Results Presentation

The data above show a varying trend based on a series of tests carried out. The number of accepted blockchain transactions on smart farming requests fell from 189,000 to 109,450 after carrying out the first three tests (phases 1 to 3). Figure 7 and Figure 8 show the testing stages and the time taken to induce the device alarm and the number of accepted blockchain-based transactions on requests as well.

However, the next three testing phases showed a rise in the number of blockchain transactions accepted on smart farming requests from 176,000 to 290,786. For real-time alert reporting, the authors [10] reported an average network latency of 0.11 s, which is cutting-edge. Our test also achieved the same latency. We further observed that the less the time taken to induce the device alarm, the higher the number of blockchain transactions accepted on smart farming requests. This demonstrates the efficacy of blockchain-based poisoning attack mitigation in smart farming. The device alarm helps minimize poisoning attacks on the blockchain network concerning smart farming requests.

## 5. Conclusions

In this study, we presented a different method to mitigate and prevent a poisoning attack in the smart farming system which notifies farmers of security and safety concerns, as well as the status of sensor devices. The end-to-end query implementation as demonstrated used an Arduino device pack, an AWS cloud environment, an Ethereum blockchain smart contract, and a web application GUI. The system can provide real-time notifications to farmers, enable remote observation of the cultivation and farming ecosystem, and connect the farming society through this smart farm blockchain-based security framework. In our approach, the execution evaluation in terms of organized idleness becomes obvious, and it can be stated that the delay can be avoided by executing powerful exchange blockchains such as Cardano. In addition, the six test results showed varying trends in the number of accepted blockchain transactions and the time taken to induce the device alarm system. The first three tests revealed a decline in the number of accepted blockchain transactions and a rise in the time taken to induce the device alarm. However, the last three tests revealed a rise in the number of accepted blockchain transactions and a fall in the time taken to induce the device alarm. The less time it takes to induce a device alarm, the higher the number of accepted blockchain transactions on smart farming requests and vice versa. This further validates the prevention of blockchain-based poisoning in smart farming and also enhances blockchain transactions and development. In the future, we intend to use a sidechain to separate the blockchain connected to the main blockchain. This can help improve scalability and performance as the sidechain can handle transactions that are not critical to the main blockchain. The data presented is a limitation of this work as we only conducted six experiments and did not fully understand the utility of a large amount of data. To analyze and predict a specific attack, one can also use neural networks and other classification approaches.

## Figures and Tables

**Figure 1 sensors-23-07992-f001:**
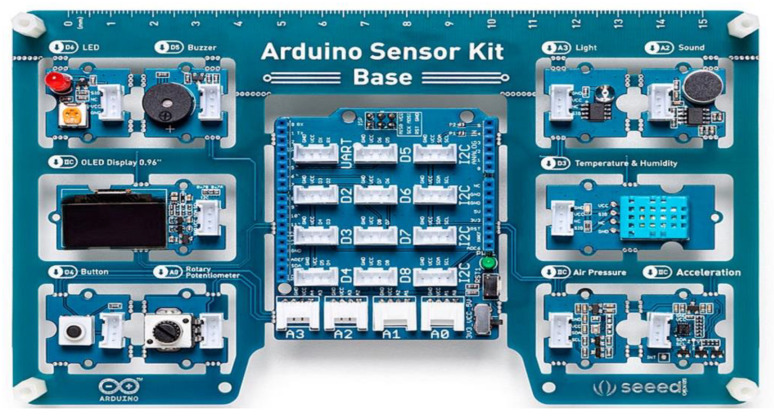
Arduino Sensor Kit [15].

**Figure 2 sensors-23-07992-f002:**
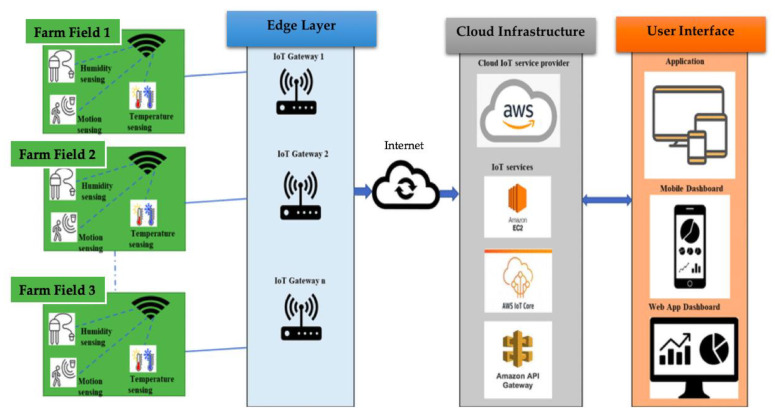
Illustration of smart farming application in cloud-based IoT [11].

**Figure 3 sensors-23-07992-f003:**
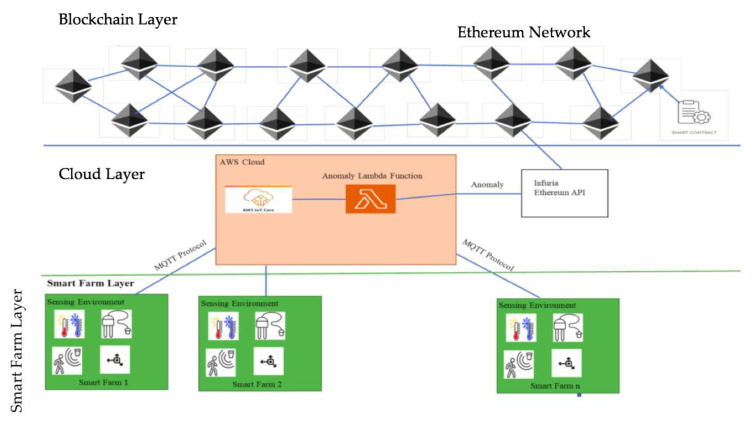
Blockchain-based solution in smart farming [10].

**Figure 4 sensors-23-07992-f004:**
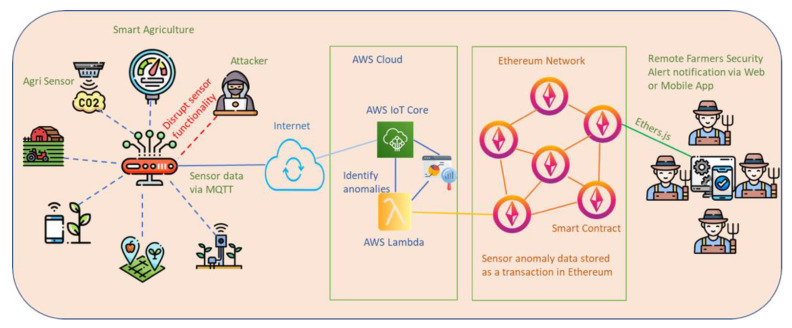
The security framework activity diagram [10].

**Figure 5 sensors-23-07992-f005:**
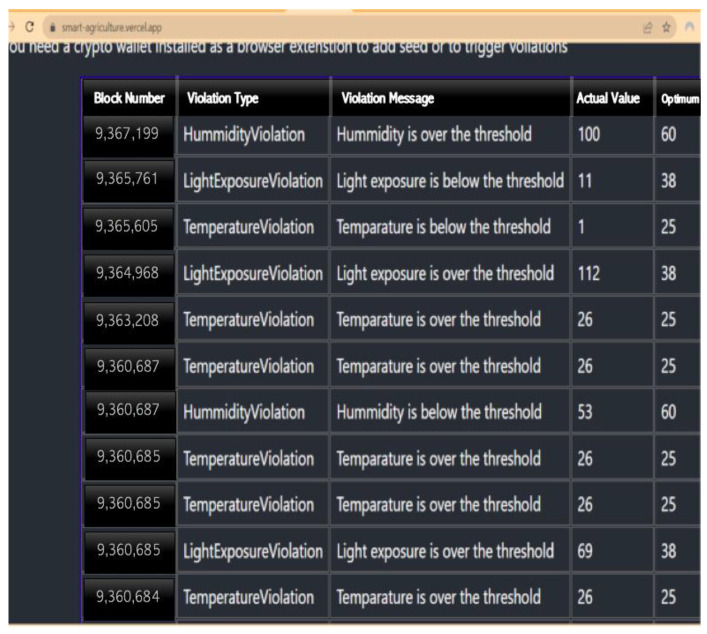
Alert for smart-contract web applications—frontend.

**Figure 6 sensors-23-07992-f006:**
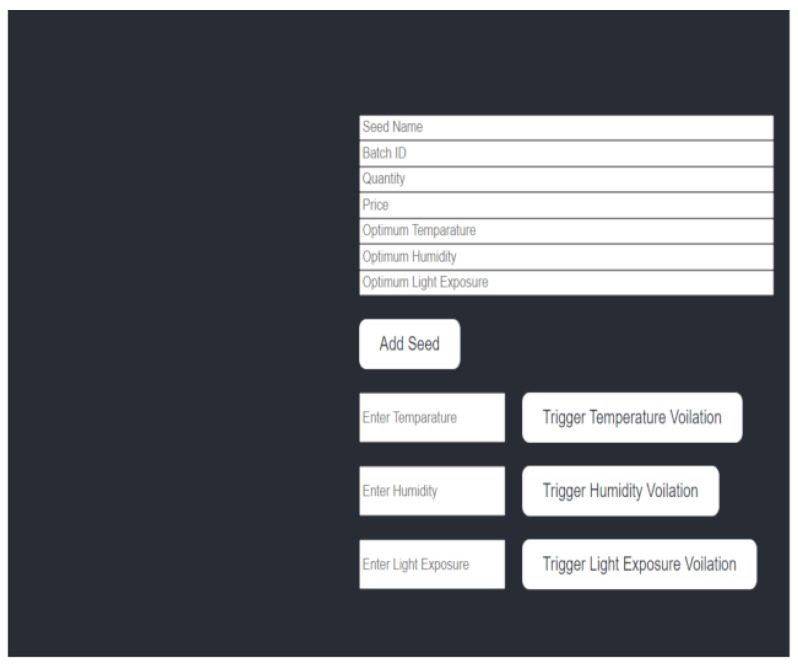
Frontend GUI for smart-contract web applications.

**Figure 7 sensors-23-07992-f007:**
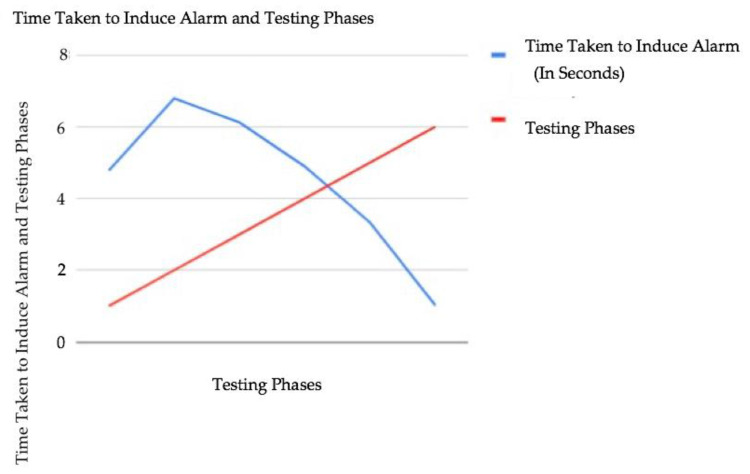
Testing stages and the time taken to induce the device alarm.

**Figure 8 sensors-23-07992-f008:**
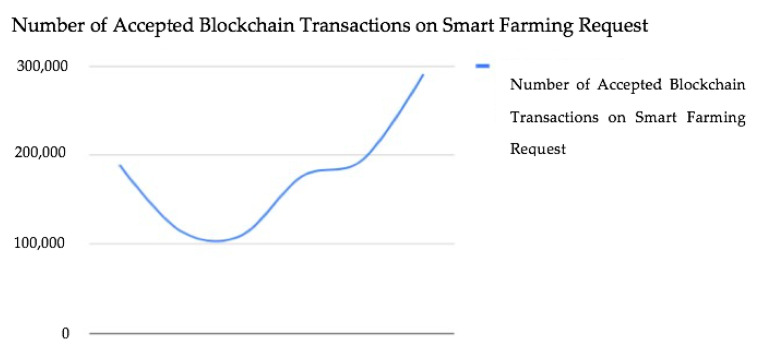
Number of accepted blockchain-based transactions on requests.

**Table 1 sensors-23-07992-t001:** Comparing related works.

Year	Authors	Main Focus/Contribution	Real-Time Response	Blockchain-Based	Review/Method
2022	[10]	A security-monitoring framework prototype for smart farms	Yes	Yes	Method
2020	[24]	Challenges of security and privacy issues in green IoT-based agriculture	No	Yes	Review
2021	[25]	A comprehensive review of emerging technologies for IoT-based smart agriculture	No	No	Review
2020	[26]	Optimization of data storage, processing, and mining of large amounts of data generated in the agricultural production process	Yes	No	Method
2016	[27]	Solutions for privacy during communication between end sensors and devices and the controller	No	No	Method
2018	[28]	A scalable network architecture for monitoring and controlling agriculture and rural areas	No	No	Method
2019	[29]	A framework for device management and resilience to attacks on the smart city network	No	Yes	Method
2022	[30]	A tamper-resistant authentication scheme for IoT devices using Constrained Application Protocol (CoAP)	No	No	Method
2023	[31]	A novel blockchain-centric IoT architecture to enable effective management of IoT data communications.	No	Yes	Method
2021	[32]	A framework for poisoning attack prevention	No	Yes	Method
2020	[33]	Novel blockchain models as solutions to major challenges in IoT-based precision agricultural systems	No	Yes	Review
2021	[34]	Examining the most recent systems that use blockchain technology to provide information security	No	Yes	Review
2023	Our Work	A framework for poisoning attack detection	Yes	Yes	Method

**Table 2 sensors-23-07992-t002:** Varying data trends.

Time Taken to Induce Device Alarm (In Seconds)	Number of Accepted Blockchain Transactions on Smart Farming Requests	Testing Phases
4.78	189,000	1
6.79	114,900	2
6.12	109,450	3
4.89	176,000	4
3.33	194,670	5
1.02	290,786	6

## Data Availability

Not Applicable.

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
