# Peer review of "Blockchain-Based Smart Farm Security Framework for the Internet of Things"

_sensors, 2023, doi:10.3390/s23187992_

Round 1

Reviewer 1 Report

The idea of this paper is current and interesting. In the era of increasing application of the Internet of Things concept in many areas, data and system security is becoming a big problem.

The introductory part is relatively well-written. It gradually introduces the reader to the area covered by this paper, and everything is covered with appropriate references. However, there are certain objections that should be corrected.

·        Line 1

"Article type" should be corrected to "Article”.

·        Lines 32, 47, 57, 95

Citations of multiple references should be in accordance with the template.

·        Line 92

Consider deleting the subtitle name “2.1. Cloud Solutions in Smart Farming” because it seems unnecessary.

The text within the subsection is fine.

·        There is no reference to Figure 1 in the text.

·        Font size of figure captions should be in accordance with the template.

·        All images should be of better quality. The size of the text in Figures 1 and 2 should be increased to make it more readable.

·        There are three figures with the caption Fig. 2

·        In section 2 there is an unclear sentence “The system Recently, Light has been shed [13] on security and safety issues in IoT as a whole and smart farming specifically where coating manufacturing and notable conceivable smart farming cyber threats were displayed.” This sentence should be rewritten to make it clearer.

·        Consider changing the name of the section "3 Materials and Method" in accordance with its content.

·        The section "3 Materials and Method" should describe the system in more detail. Authors should refer to the AWS Lambda functionality as a central part of the system's operation and describe its functionality in more detail, for example through a description of the operation in the form of an algorithm.

·        There is no reference to Table 1 in the text.

·        The biggest objection is to section “4 Results Discussion”. This section should be further elaborated and clarified.

o    There is an unclear sentence “The data were obtained from the phase of triggering the device alarm when the organized sleep is in seconds, to verify the number of blockchain transactions accepted 187 based on smart farming requests.” This sentence should be rewritten to make it clearer.

o    The text states that the testing was carried out in six phases. What do these testing phases represent? How are these phases of testing different? How long did each phase of testing take?

o    In Table 1 what is the “Time Taken to Induce Device parameter Alarm (In seconds)”? Are the average values of this parameter at the end of each testing phase given in Table 1? Or these are the values that the testers set for each phase for the purposes of system testing? If these are average values, why was there such a difference in the different phases of the test? If these are the values set by the testers, why were these values chosen?

·        In conclusion, the authors stated, "The end-to-end query implementation as demonstrated used an Arduino device pack, an AWS cloud environment, Ethereum block-chain smart contract, and a web application GUI. The system can provide real-time notifications to farmers, enable remote observation of the cultivation and farming ecosystem, and connect the farming society through this smart farm blockchain-based security framework." This statement is not substantiated in the text. The suggestion to the authors is to include screenshots of functional applications with examples of the user interface and notifications sent to farmers, or to post a web link with a demo account where the functionality of the system can be tested.

·        In conclusion,  statement “The first three tests revealed a decline in the number of accepted Blockchain  transactions and a rise in the time taken to induce device alarm. However, the last three tests revealed a rise in the number of accepted blockchain transactions and a fall in the  time taken to induce device alarm. The lesser time it takes to induce a device alarm, the higher the number of accepted blockchain transactions on smart farming requests and vice versa. This further validates the prevention of blockchain based poisoning in smart farming and also enhances Blockchain transactions and development.”  cannot be confirmed until section 4 is explained in more detail and clarity.

Reviewer 2 Report

- The authors should include numerical restlts obtained in teh abstract.

- The authors should proofread the paper for grammatical mistakes. For example, "Normally, As the population of the world grow......" should have been "Normally, as the population of the world grow.....".

- The innovation/novelty of this work has to be discussed in detail in teh introduction.

- What are the key contributions of this study?

- Summarize the findings from the recent works in the form of a table.

- Some of the recent works such as the following can be discussed SP2F: A secured privacy-preserving framework for smart agricultural Unmanned Aerial Vehicles

- The results section is very weak. Teh authors have to significantly enhance the results section. They can add more analysis on teh results such as computational complexity of the proposed blockchain based approach, etc.

- The authors should compare the results obtained with recentstate of the art.

- A detailed analysis that includes the inferences of the authors on the reults obtained has to be presented.

- Discuss in detail about the limitations of this work along with possible future enhancements.

The author shav eto proofread teh article carefully.

Reviewer 3 Report

The paper presents an approach to bolstering security in smart farming systems by leveraging the Ethereum blockchain to track smart contracts and detect discrepancies in security checks. Employing a series of IoT devices, the study utilizes the AWS cloud environment, including AWS Lambda functions, to analyze data and interface with the Ethereum blockchain through Infura API. Tests were conducted to gauge the efficacy of this system in mitigating blockchain-based poisoning attacks, with findings indicating a correlation between the time taken to trigger device alarms and the number of accepted blockchain transactions. The framework offers real-time notifications for farmers, remote monitoring capabilities, and an enhanced security structure, while also highlighting potential areas for future improvement, such as the integration of more efficient blockchains like Cardano.

Major comments:

1. Several sections of the text appear truncated or lack sufficient context, potentially leaving readers without a complete understanding, especially if they're not well-versed in the subject matter.

2. The paper delves into the use of the Ethereum blockchain and briefly mentions alternatives like Cardano. However, a detailed comparative analysis of various blockchain technologies, specifically in the realm of smart farming, is missing.

3. The described methodology could benefit from clearer explanations regarding the interactions between its components, especially for readers unfamiliar with AWS Lambda, AWS IoT core, and Ethereum.

4. There seems to be a strong technical focus in the paper, with insufficient discussion about the broader, real-world implications of the findings within agricultural contexts.

5. While the literature review is thorough in certain aspects, a more comprehensive exploration of diverse sources and previous studies tackling similar challenges could enrich it.

6. The rationale behind conducting six testing phases is unclear. A detailed breakdown highlighting the distinct objectives and variations between these phases would be beneficial.

7. The discourse on poisoning attacks and their mitigation is somewhat surface-level. A deeper investigation into potential security threats or vulnerabilities, especially those inherent to smart farming systems, might enhance the paper's depth.

8. Some conclusions seem to generalize specific test outcomes. It would be essential to contextualize these findings to determine their broader applicability.

9. Referencing an "Arduino device pack" without specifying its particular model or configuration could hinder replication efforts or the assessment of its relevance to diverse setups.

10. Certain terminologies, such as "organized sleep", "end-to-end query implementation", or "organized idleness", could be clarified or expanded upon to enhance their universal comprehension.

11. The scalability concerns of blockchain solutions, particularly those based on Ethereum, are not addressed. Discussing how the proposed solution might fare in a large-scale implementation could be crucial for its practicality.

There are instances where the sentence structure and wording can be refined for better clarity and coherence.

Round 2

Reviewer 1 Report

·        Multiple references needУ to be matched to the template.

In the paper: [2] [3]   

In the template: References should be numbered in order of appearance and indicated by a numeral or numerals in square brackets—e.g., [1] or [2,3], or [4–6].

There are errors in multiple places in the paper that must be corrected.

·        It is necessary to put a space between words and references.

In paper:   devices[6].

It is necessary to check the entire text and correct errors

·        Line 77:   (in figure 1)   change to (in Figure 1)

·        Lines 80 – 95:  Strange text next to Figure 1

·        Line 211: The spacing between paragraphs should be in accordance with the template.

·        Table borders should be in accordance with the template. It is necessary to change the appearance of the border on all tables.

·        Line 245: It is necessary to add the section number.

·        Line 285: The sentence starts with a lowercase letter.

·        Line 327: It is necessary to add the section number.

·        Line 354: web link don’t work. 404: NOT_FOUND Code: DEPLOYMENT_NOT_FOUND

ID: fra1::j8d6f-1692603049387-1d70ff517717

·        Line 406: The sentence starts with a lowercase letter and it is unclear.

“g the device alarm when…”

·        Table borders should be in accordance with the template. It is necessary to change the appearance of the border on all tables.

·        Line 245: It is necessary to add the section number.

·        Line 285: The sentence starts with a lowercase letter.

·        Line 327: It is necessary to add the section number.

·        Line 354: web link don’t work. 404: NOT_FOUND Code: DEPLOYMENT_NOT_FOUND

ID: fra1::j8d6f-1692603049387-1d70ff517717

·        Line 406: The sentence starts with a lowercase letter and it is unclear.

“g the device alarm when…”

Reviewer 2 Report

All the comments are addressed

Reviewer 3 Report

Thanks for addressing my comments.
